# Active Learning Through a Covering Lens

**Ofer Yehuda**[†]**, Avihu Dekel**[†]**, Guy Hacohen**[†‡]**, Daphna Weinshall**[†]
School of Computer Science & Engineering[†]
Edmond and Lily Safra Center for Brain Sciences[‡]
The Hebrew University of Jerusalem
Jerusalem 91904, Israel
{ofer.yehuda,avihu.dekel,guy.hacohen,daphna}@mail.huji.ac.il

## Abstract

Deep active learning aims to reduce the annotation cost for the training of deep models, which is notoriously data-hungry. Until recently, deep active learning methods were ineffectual in the *low-budget* regime, where only a small number of examples are annotated. The situation has been alleviated by recent advances in representation and self-supervised learning, which impart the geometry of the data representation with rich information about the points. Taking advantage of this progress, we study the problem of subset selection for annotation through a "covering" lens, proposing *ProbCover* – a new active learning algorithm for the low budget regime, which seeks to maximize *Probability Coverage*. We then describe a dual way to view the proposed formulation, from which one can derive strategies suitable for the high budget regime of active learning, related to existing methods like *Coreset*. We conclude with extensive experiments, evaluating *ProbCover* in the low-budget regime. We show that our principled active learning strategy improves the state-of-the-art in the low-budget regime in several image recognition benchmarks. This method is especially beneficial in the semi-supervised setting, allowing state-of-the-art semi-supervised methods to match the performance of fully supervised methods, while using much fewer labels nonetheless. Code is available at https://github.com/avihu111/TypiClust.

## 1   Introduction

For the most part, deep learning technology critically depends on access to large amounts of annotated data. Yet annotations are costly and remain so even in our era of *Big Data*. Deep active learning (AL) aims to alleviate this problem by improving the utility of the annotated data. Specifically, given a fixed budget $b$ of examples that can be annotated, and some deep learner, AL algorithms aim to query those $b$ examples that will most benefit this learner.

In order to optimally choose unlabeled examples to be annotated, most deep AL strategies follow some combination of two main principles: 1) Uncertainty sampling [e.g., 26, 48, 3, 14, 15, 36, 25], in which examples that the learner is most uncertain about are picked, to maximize the added value of the new annotations. 2) Diversity Sampling [e.g., 1, 22, 16, 38, 18, 40, 37, 17, 47, 42, 46], in which examples are chosen from diverse regions of the data distribution, to represent it wholly and reduce redundancy in the annotation.

Most AL methods fail to improve over random selection when the annotation budget is very small [35, 39, 7, 32, 55, 21, 2], a phenomenon sometimes termed "cold start" [8, 49, 16, 50, 23]. When the budget contains only a few examples, they struggle to improve the model's performance, and even fail to reach the accuracy of the random baseline. Recently, it was shown that uncertainty sampling is inherently unsuited for the low-budget regime, which may explain the cold start phenomenon [19]. The low-budget scenario is relevant in many applications, especially those requiring an expert tagger

36th Conference on Neural Information Processing Systems (NeurIPS 2022).

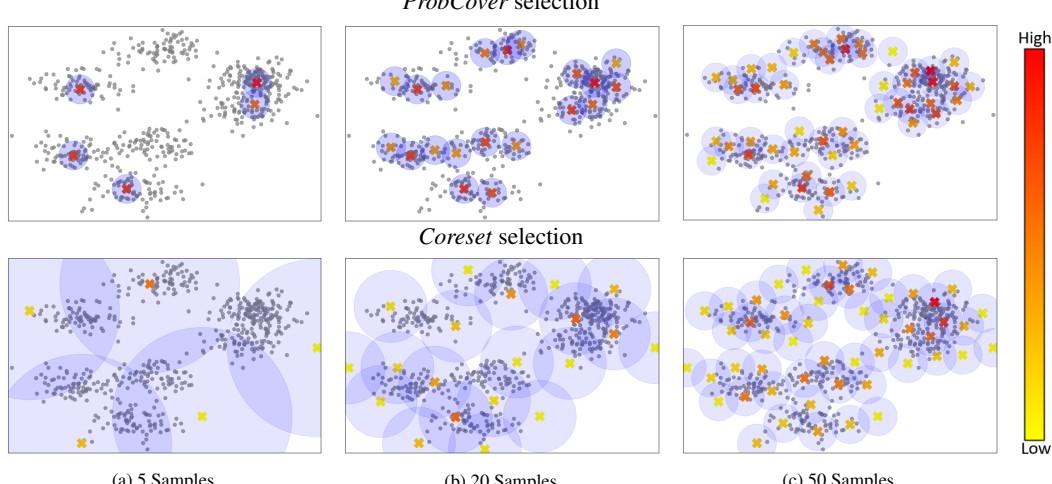

Figure 1: *ProbCover* selection (top) vs *Coreset* selection (bottom) of 5/20/50 samples (out of 600). Selected points are marked by **x**, which is color-coded by density (see color code bar to the right). Density is measured using Gaussian Kernel Density Estimation, and the covered area is marked in light blue. *Coreset* attempts to minimize ball size, constrained by complete coverage, while *ProbCover* attempts to maximize coverage, constrained by a fixed ball size. Note that especially in low budgets, such as 5 samples, *Coreset* only selects outliers of the distribution (yellow), while *ProbCover* selects from dense regions of the distribution (red).

whose time is expensive (e.g., a radiologist tagger for tumor detection). If we want to expand deep learning to new domains, overcoming the cold start problem is an ever-important task.

In this work, we focus on understanding the very low budget regime of AL, where the budget of $b$ examples cannot dependably represent the annotated data distribution. To face up to this challenge, in Sections 2.1-2.2 we model the problem as *Max Probability Cover*, defined as follows: given some data distribution, and a radius $\delta$, select the $b$ examples that maximize the probability of the union of balls with radius $\delta$ around each example. We further show that under a separation assumption that is realistic in semantic embedding spaces, *Max Probability Cover* is befitting the nearest-neighbor classification model, in that it minimizes an upper bound on its generalization error.

In Section 2.4 we show a connection with existing deep AL methods, like *Coreset* [37], and explain why those methods are more suitable for the high-budget regime than the low-budget regime. This phenomenon is visualized in Fig. 1, where we see that with only a few examples to choose, *Coreset* – an AL strategy that employs the principle of diversity sampling – chooses distant and often abnormal points, while *ProbCover* chooses representative examples.

When using the empirical data distribution, we further show that *Max Probability Cover* can be reduced to *Max Coverage* – a known classical NP-hard problem [34] (see Section 2.2). To obtain a practical AL strategy, in Section 3 we adapt a greedy algorithm for the selection of $b$ examples from a fixed finite training set of unlabeled examples (the training set), which guarantees $1 - \frac{1}{e}$ approximation to the problem. We call this new method *ProbCover*.

In Section 4 we empirically evaluate the performance of *ProbCover* on several computer vision datasets, including CIFAR-10, CIFAR-100, Tiny-ImageNet, ImageNet and its subsets. *ProbCover* is thus shown to significantly outperform all alternative deep AL methods in the very low-budget regime. Additionally, *ProbCover* improves the performance of state-of-the-art semi-supervised methods, which were thought until recently to make AL redundant [6], allowing for the learning of computer vision tasks with very few annotated examples.

**Relation to prior art** Recent work investigated AL methods based on an approximation to a *facility location problem* [45, 30, 54, 37, 43], which is a variant of the covering problem. In the *minimax facility location problem* [13], the entire distribution is covered with a fixed number of balls, which can vary in size, whereas in *ProbCover* the size of the balls is fixed, and we are allowed to cover only part of the total distribution. While this difference may seem minor, in the low-budget regime, when the budget is not large enough to represent the data, examples chosen by the facility location problem are not representative (as illustrated in Fig. 1), which leads to poor performance (as shown in Fig. 10).

**Summary of contribution**

    (i) Develop a theoretical framework to analyze AL strategies in embedding spaces, with "dual" low and high-budget interpretations.

    (ii) Introduce *ProbCover*, a low-budget AL strategy motivated by our framework, which significantly outperforms other methods in the low-budget regime.

    (iii) Demonstrate the outstanding competence of *ProbCover* in semi-supervised learning with very few labeled examples.

## 2 Theoretical Analysis

To address the challenge of active learning in low budgets, we adopt a point coverage framework. Computationally, we analyze the generalization of the Nearest Neighbor (NN) classification model, as this model depends exclusively on distances from a set of training examples and does not involve any additional inductive bias. Thus in Section 2.1 we develop a bound on the generalization error in 1-NN models. It follows from the analysis (see discussion below) that if full coverage is required, which is only practical in the high budget regime, the minimization of this bound translates to the *optimal minimax facility location* problem, which is known to be NP-hard and which the AL *Coreset* algorithm by Sener and Savarese [37] is designed to approximate.

In contrast, in the low budget regime, the aforementioned bound can best be optimized by seeking a labeled set $L$ whose probability of covering the unlabeled set is maximal. In Section 2.2 we show that this problem is also NP-hard. Furthermore, when the data distribution is not known and is being approximated by the empirical distribution, we show that it is equivalent to the classical *Max Coverage* problem. *ProbCover* described in Section 3, is designed to solve this problem. In Section 2.4 we discuss a sense in which the high budget and low budget problems are dual.

### 2.1 Bounding the Generalization Error

We shall now derive a bound on the generalization error of the 1 Nearest Neighbor (1-NN) classifier. We start with some necessary notations and definitions. Most important is the assumption of $\delta$-purity, which states that most of the time, points that are less than $\delta$ apart have the same label. We then prove a lemma, showing that given a labeled set $L$ and the coverage it achieves, and given the $\delta$-purity assumption, the probability of a point being inside this cover and still being falsely labeled is small. From this, we finally derive a bound on the generalization error, which is stated in Thm. 1.

**Notations** Let $\mathbb{X}$ denote the input domain whose underlying probability function is denoted $P$, and let $\mathbb{Y} = [k]$ denote the target domain. Assume that a true labeling function $f : \mathbb{X} \to \mathbb{Y}$ exists. Let $X = \{x_i\}_{i=1}^{m}$ denote an unlabeled set of points, and $b \leq m$ the annotation budget. Let $L \subseteq X$ denote the labeled set, where $|L| = b$. Let $B_\delta(x) = \{x' : \|x' - x\|_2 \leq \delta\}$ denote a ball centered at $x$ of radius $\delta$. Let $C \equiv C(L, \delta) = \bigcup_{x \in L} B_\delta(x)$ denote the region covered by $\delta$-balls centered at the labeled examples in $L$. We call $C(L, \delta)$ the *covered region* and $P(C)$ the *coverage*.

**Definition 2.1.** We say that a ball $B_\delta(x)$ is *pure* if $\forall x' \in B_\delta(x) : f(x') = f(x)$.

**Definition 2.2.** We define the *purity* of $\delta$ as

$$\pi(\delta) = P(\{x : B_\delta(x) \text{ is pure}\}).$$

Notice that $\pi(\delta)$ is monotonically decreasing.

Let $\hat{f}$ denote the 1-NN classifier based on $L$. We split the covered region $C(L, \delta)$ into two sets:

$$C_{right} = \{x \in C : \hat{f}(x) = f(x)\}, \qquad C_{wrong} = C \setminus C_{right}.$$

**Lemma 1.** $C_{wrong} \subseteq \{x : B_\delta(x) \text{ is not pure}\}$.

*Proof.* Let $x \in C_{wrong}$. Let $c \in L$ denote the nearest neighbor to $x$. Then they have the same predicted label, $\hat{f}(x) = \hat{f}(c)$, and $f(c) = \hat{f}(c)$ because $c$ is labeled. Since $x$ is wrongly labeled, $\hat{f}(x) \neq f(x)$, which implies that

$$f(c) = \hat{f}(c) = \hat{f}(x) \neq f(x).$$

Finally, since $x \in C_{wrong} \subseteq C$ is in the coverage, $d(x, c) < \delta$, which means that $c \in B_\delta(x)$ with a different label and so $B_\delta(x)$ is not pure. $\square$

**Corollary 1.**
$$P(C_{wrong}) \leq P(\{x : B_\delta(x) \text{ is not pure}\}) = 1 - \pi(\delta).$$

**Theorem 1.** The generalization error of the 1-NN classifier $\hat{f}$ is bounded as follows

$$\mathbb{E}\left[\hat{f}(x) \neq f(x)\right] \leq (1 - P(C(L, \delta))) + (1 - \pi(\delta)). \tag{1}$$

*Proof.*

$$
\begin{aligned}
\mathbb{E}\left[\hat{f}(x) \neq f(x)\right] &= \mathbb{E}[\mathbb{1}_{f(x) \neq \hat{f}(x)} \mathbb{1}_{x \notin C}] + \mathbb{E}[\mathbb{1}_{f(x) \neq \hat{f}(x)} \mathbb{1}_{x \in C}] \\
&\leq P(x \notin C) + \mathbb{E}[\mathbb{1}_{f \neq \hat{f}} \mathbb{1}_{x \in C_{right}}] + \mathbb{E}[\mathbb{1}_{f(x) \neq \hat{f}(x)} \mathbb{1}_{x \in C_{wrong}}] \\
&\leq P(x \notin C) + 0 + P(x \in C_{wrong}) \\
&\leq (1 - P(C(L, \delta))) + (1 - \pi(\delta)). \qquad \square
\end{aligned}
$$

Note that (1) gives us a different bound for different $\delta$ values, which also depends on the labeled set $L$. This bound introduces a trade-off: as $\delta$ increases, the *coverage* increases, but the *purity* decreases. Ideally, we should seek a pair $\{\delta, L\}$ that achieves the tightest bound.

**Discussion**  We can interpret (1) in the context of two boundary conditions of AL: high-budget and low-budget. In the high-budget regime, achieving full coverage $P(C) = 1$ is feasible as we have many points, and the remaining challenge is to reduce $1 - \pi(\delta)$. Accordingly, since $\pi(\delta)$ is monotonically decreasing, we seek to minimize $\delta$ subject to the constraint $P(C) = 1$. This is similar to *Coreset* [37]. In the low-budget regime, full coverage entails very low purity, which (if sufficiently low) makes the bound trivially 1. Thus, instead of insisting on full coverage, we fix a $\delta$ that yields "large enough" purity $\pi(\delta) > 0$, and then seek a labeled set $L$ that maximizes the coverage $P(C)$. We call this problem *Max Probability Cover*.

## 2.2   Max Probability Cover

**Definition 2.3** (*Max Probability Cover*)**.** Fix $\delta > 0$, and obtain a subset $L \subset X$, $|L| = b$, that maximizes the probability of the covered area $P(C(L, \delta))$

$$\underset{L \subseteq X; |L| = b}{\operatorname{argmax}} \ P(\bigcup_{x \in L} B_\delta(x)) \tag{2}$$

An optimal solution to (2) would minimize the bound in (1), when $\delta$ is fixed.

Unfortunately, when moving to practical settings there are two obstacles. The first is complexity:

**Theorem 2.** *Max Probability Cover* is NP-hard.

*Proof.* (Sketch, see full proof in App. B) We construct a reduction from an established NP-hard problem (*Max Coverage*, see Def. A.1) to *Max Probability Cover*. For the collection of subsets $S = \{S_1, \ldots, S_m\}$, we consider the space $\mathbb{R}^m$ and a collection of $\delta$-balls $\{B_\delta(x_i)\}_{i=1}^m$ with the *exhaustive intersection* property. This means that any subset of the balls has at least one point that is contained in all the balls in the subset, but not contained in any other ball (see example in Fig. 2). The existence of such a collection of balls in $\mathbb{R}^m$, $\forall m$, is proved in Lemma 3 (see App. B). We then assign each $S_i$ to $B_\delta(x_i)$, and each element in $S_i$ is mapped to a point in the intersection of all the balls assigned to subsets that contain it. Each such point then defines a Dirac measure, the normalized sum of which determines a probability distribution on $\mathbb{R}^m$. The selection of $\delta$-balls that is the solution to the *Max Probability Cover* can be translated back to a selection of subsets, which is the solution to the original *Max Coverage* problem. $\square$

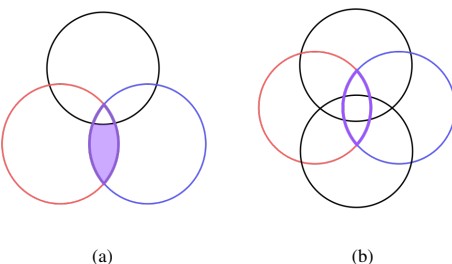

(a)                                    (b)

Figure 2: Illustration in $\mathbb{R}^2$ of *exhaustive intersection* (see Def. B.2). (a) With 3 balls, every subset of balls has a point that is contained *only* in the specific subset. Thus this set of 3 balls has the *exhaustive intersection* property. (b) With 4 balls, any point in the intersection of two opposite balls is also contained in at least one other ball. Thus this set of 4 balls does **not** have the *exhaustive intersection* property. In the drawing, the region of intersection between the red and blue balls is outlined in purple, while the points within the region that are unique to this pair are marked in light purple. Note that in example (b), this set is empty.

## 2.3   Using the Empirical Distribution

When employing *Max Probability Cover*, the second practical problem concerns the data distribution, which is hardly ever known apriori. In fact, even when known, the subsequent probabilistic computations are often intractable and hard to approximate. Instead, we may use the *empirical distribution* $\tilde{P}(A) = \frac{1}{m} \sum_{i=1}^{m} \mathbb{1}_{x_i \in A}$ as an approximation, which gives us the following useful result:

**Proposition.** When $P$ is the empirical distribution $\tilde{P}$, the *Max Probability Cover* objective function is equivalent to the *Max Coverage* objective, with $\{B_\delta(x_i) \cap X\}_{i=1}^{m}$ as the collection of subsets.

*Proof.* Given a labeled set $L = \{x_i\}_{i=1}^{b}$, we show equality of objectives up to constant $\frac{1}{|X|}$

$$
P\left(\bigcup_{i=1}^{b} B_\delta(x_i)\right) = P\left(\{y \in \mathbb{R}^d \mid \exists i \quad \|x_i - y\| < \delta\}\right)
$$

$$
= \frac{1}{|X|} |\{x \in X \mid \exists i \quad \|x_i - x\| < \delta\}|
$$

$$
= \frac{1}{|X|} \left|\bigcup_{i=1}^{b} (B_\delta(x_i) \cap X)\right|
$$

$\square$

## 2.4   The "Duality" of *Max Probability Cover* and *Coreset*

The *Coreset* AL method by Sener and Savarese [37] minimizes the objective

$$
\delta(L) = \max_{x \in X} \min_{c \in L} d(x, c) = \min\{\delta \in \mathbb{R}_+ : X \subseteq \bigcup_{c \in L} B_\delta(c)\}
$$

We can rewrite the above in the language of distributions as

$$
\delta'(L) = \min\{\delta \in \mathbb{R}_+ : P(\bigcup_{c \in L} B_\delta(c)) = 1\}
$$

If we use the empirical distribution then $\delta(L) = \delta'(L)$. In this framework we can say that *Max Probability Cover* and *Coreset* are dual problems in the following loose sense:

1. *Max Probability Cover* minimizes the generalization error bound (1) when we fix $\delta$ and seek to maximize the coverage, which is suitable for the low budget regime.

2. *Coreset* minimizes the generalization error bound (1) when we fix the coverage to 1 and minimize $\delta$, which is suitable for the high budget regime because only then can we fix the coverage to 1.

This duality is visualized in Fig. 1.

# 3   Method: *ProbCover*

To deliver a practical method, we first note that our approach implicitly relies on the existence of a good embedding space [4, 10, 53], where distance is correlated with semantic similarity, and where similar points are likely to bunch together in high-density regions. As is now customary [e.g., 30, 19], we use an embedding space derived by training a self-supervised task over the large unlabeled pool. In such a space similar labels often correspond to short distances, making 1-NN classification suitable, and also providing for the existence of large enough $\delta$ balls with good purity and coverage properties.

Secondly, we note that *Max Coverage* is NP-hard and cannot be solved efficiently. Instead, as its objective is submodular and monotone [27], we use the greedy approximate algorithm that achieves $\left(1 - \frac{1}{e}\right)$-approximation [27]. A better approximation is impractical, as shown in App. D.1. See App. E for additional time and space complexity analysis.

Below, we describe the greedy algorithm in Section 3.1, and the estimation of ball size $\delta$ in Section 3.2.

## 3.1   Greedy Algorithm

---
**Algorithm 1** *ProbCover*

---
**Input:** unlabeled pool $U$, labeled pool $L$, budget $b$, ball-size $\delta$,
**Output:** a set of points to query
$X \leftarrow$ Embedding of representation learning algorithm on $U \cup L$
$G = (V = X, E = \{(x, x') : x' \in B_\delta(x)\})$
**for all** $c \in L$ **do**
   Remove the incoming edges to covered vertices, $\{(x', x) \in E : (c, x) \in E\}$, from $E$
**end for**
*Queries* $\leftarrow \emptyset$
**for all** i=1,...,b **do**
   Add $c \in U$ with the highest out-degree in $G$ to *Queries*
   Remove the incoming edges to covered vertices, $\{(x', x) \in E : (c, x) \in E\}$, from $E$
**end for**
**return** *Queries*

---

The algorithm (see Alg. 1 below for pseudo-code) goes as follows: First, construct a directed graph $G = (V, E)$, with $V = X$ the embedding of the data space, and $(x, x') \in E \iff x' \in B_\delta(x) \iff d(x, x') \leq \delta$. In $G$, each vertex represents a specific example, and there is an edge between two vertices $(x, x')$ if $x'$ is covered by the $\delta$-ball centered at $x$ (distances are measured in the embedding space). The algorithm then performs $b$ iterations of the following two steps:

(i) Pick the vertex $x_{max}$ with the highest out-degree for annotation;

(ii) Remove all incoming edges to $x_{max}$ and its neighbors.

As *ProbCover* uses a sparse representation of the adjacency graph, it is able to scale to large datasets while requiring limited space resources. The complexity analysis of the algorithm, and specifically the complexity of constructing the *adjacency graph* and of the *sample selection*, are discussed in the Appendix E.

## 3.2   Estimating $\delta$

Our algorithm requires the specification of hyper-parameter $\delta$, the ball radius, whose value depends on details of the embedding space (see App. C.1 for embeddings used). In choosing $\delta$, we need to consider the trade-off between large coverage $P(C)$ and high purity $\pi(\delta)$. We resolve this trade-off with the following heuristic, where we pick the largest $\delta$ possible, while maintaining purity above a certain threshold $\alpha \in (0, 1)$. Specifically,

$$\delta^* = \max\{\delta : \pi(\delta) \geq \alpha\}$$

Importantly, $\alpha$ is more intuitive to tune, and is kept constant across different datasets (unlike $\delta$). We still need to estimate the purity $\pi(\delta)$, which depends on the labels, from unlabeled data. To this end, we estimate purity using unsupervised representation learning and clustering. First, we cluster

self-supervised features using $k$-means with $k$ equal to the number of classes. For a given $\delta$, we compute the purity $\pi(\delta)$ using the clustering labels as pseudo-labels for each example. Searching for the best $\delta$, we repeat the process and pick the largest $\delta$ so that at least $\alpha = 0.95$ of the balls are pure.

In Fig. 3, we plot the percentage of pure balls across different datasets as a function of $\delta$, where the dashed line represents the $\delta^*$ chosen by *ProbCover*.

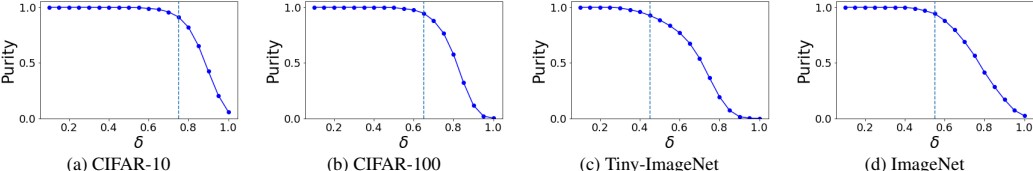

Figure 3: Ball purity, as a function of $\delta$, estimated from the unlabeled data (see text). The dashed line marks the highest $\delta$, after which purity drops below $\alpha = 0.95$.

## 4 Empirical Results

We report a set of empirical results, comparing *ProbCover* to other AL strategies in a variety of settings. We focus on the very low budget regime, with a budget size $b$ of a similar order of magnitude as the number of classes. Note that since the data is picked from an unlabeled pool, chances are that the initial labeled set is not going to be balanced across classes, and in the early stages of training some classes will almost always be missing. *ProbCover*'s excellent performance nevertheless, as seen below, demonstrates its robustness in the presence of this hurdle.

### 4.1 Methodology

Three deep AL frameworks are evaluated:

(i) *Fully supervised*: train a ResNet-18 only on the annotated data, as a fully supervised task.

(ii) *Semi-supervised by transfer learning*: create a representation of the data by training with a self-supervised task on the unlabeled data, then construct a 1-NN classifier using the ensuing representation in a supervised manner. This framework is intended to capture the basic benefits of semi-supervised learning, regardless of the added benefits provided by modern semi-supervised learning methods and the more sophisticated derivation of pseudo-labels.

(iii) *Fully semi-supervised*: train a competitive semi-supervised model on both the annotated and unlabeled data. In our experiments we use FlexMatch by Zhang et al. [51].

In frameworks (i) and (ii) we adopt the evaluation kit created by Munjal et al. [33], in which we can compare multiple deep AL strategies in a principled way. In framework (iii), we adopt the code and hyper-parameters provided by FlexMatch.

When evaluating frameworks (i) and (ii), we compare *ProbCover* to 9 deep AL strategies as baselines. (1) **Random** query uniformly. (2)-(4) query examples with the lowest score, using the following basic scores: (2) **Uncertainty** – max softmax output, (3) **Margin** – margin between the two highest softmax outputs, (4) **Entropy** – inverse entropy of softmax outputs. (5) **BADGE** [1]. (6) **DBAL** [15]. (7) **TypiClust** [19]. (8) **BALD** [26]. (9) **W-Dist** [30], see also App. D.4. (10) **Coreset** [37]. We note that while most baseline methods are suitable for the high budget regime, *TypiClust* and *W-Dist* are also suitable for the low budget regime. Similarly to *ProbCover*, *TypiClust* requires a good embedding space to work properly. When comparing *ProbCover* and *TypiClust*, and in order to avoid possible confounds, we use the same embedding space for both methods.

These AL methods are evaluated on the following classification datasets: CIFAR-10/100 [28], TinyImageNet, [29], ImageNet [12] and its subsets (following Van Gansbeke et al. [41]). When considering CIFAR-10/100 and TinyImageNet, we use as input the embedding of SimCLR [9] across all methods. When considering ImageNet we use as input the embedding of DINO [5] throughout. Results on ImageNet-50/100 are deferred to App. D.1. Details concerning specific networks and hyper-parameters can be found in App. C, and in the attached code in the supplementary material. When evaluating frameworks (i) and (ii), we perform 5 active learning rounds, querying a fixed

budget of $b$ examples in each round. In framework (iii), as FlexMatch is computationally demanding, we only evaluate methods on their initial pool selection capabilities.

## 4.2 Main Results

**(i) Fully supervised framework.** We evaluate different AL methods based on the performance of a deep neural network trained directly on the raw queried data. In each round, we query $b$ samples where $b$ is equal to the number of classes in each dataset, and train a ResNet-18 on the accumulated queried set. We repeat this for 5 active learning rounds, and plot the mean accuracy of 5 repetitions (3 for ImageNet) in Fig. 4 (see App. D.1 for additional results).

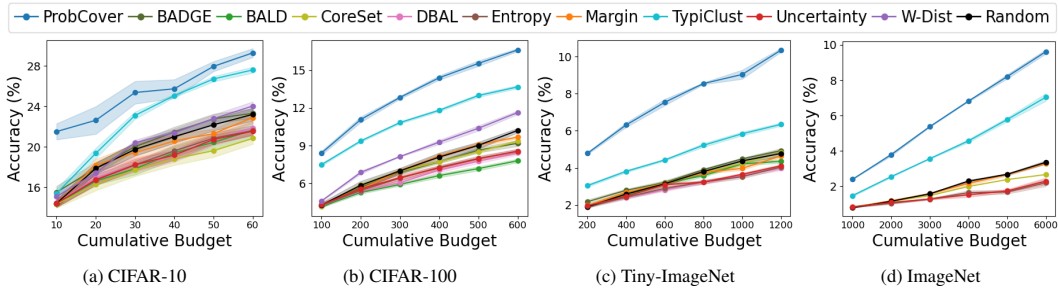

(a) CIFAR-10     (b) CIFAR-100     (c) Tiny-ImageNet     (d) ImageNet

Figure 4: Framework (i), fully supervised: The performance of *ProbCover* is compared with baseline AL strategies in image classification tasks in the low budget regime. Budget $b$ guarantees on average 1 sample per class, thus the initial sample may be imbalanced. The final average test accuracy in each iteration is reported, using 5 repetitions (3 for ImageNet). The shaded area reflects the standard error across repetitions.

**(ii) Semi-supervised by transfer learning.** In this framework, we make use of pretrained self-supervised features, and measure classification performance using the 1-NN classifier. Accordingly, each point is classified by the label of its nearest neighbor (within the selected labeled set $L$) in the self-supervised features space. In low budgets, this framework outperforms the fully-supervised framework (i), though it is not as effective as the full-blown semi-supervised learning framework (iii). This supports the generality of our findings, not limited to any specific semi-supervised method. Similarly to Fig. 4, in Fig. 5 we plot the mean accuracy of 5 repetitions for the different tasks.

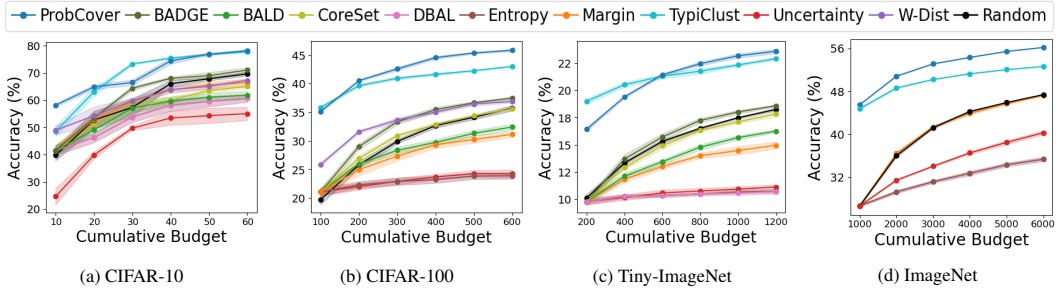

(a) CIFAR-10     (b) CIFAR-100     (c) Tiny-ImageNet     (d) ImageNet

Figure 5: Comparative evaluation of framework (ii) - semi-supervised by transfer learning, see caption of Fig. 4.

**(iii) Semi-supervised framework.** We compare the performance of different AL strategies used prior to running FlexMatch, a state-of-the-art semi-supervised method. In Fig. 6 we show results with 3 repetitions of FlexMatch, using the labeled sets provided by different AL strategies and budget $b$ equal to the number of classes. We see that *ProbCover* outperforms random sampling and other AL baselines by a large margin. We note that in agreement with previous works [6, 19], AL strategies that are suited for high budgets do not improve the results of random sampling, while AL strategies that are suited for low budgets achieve large improvements.

## 4.3 Ablation Study

We report a set of ablation studies, evaluating the added value of each step of *ProbCover*.

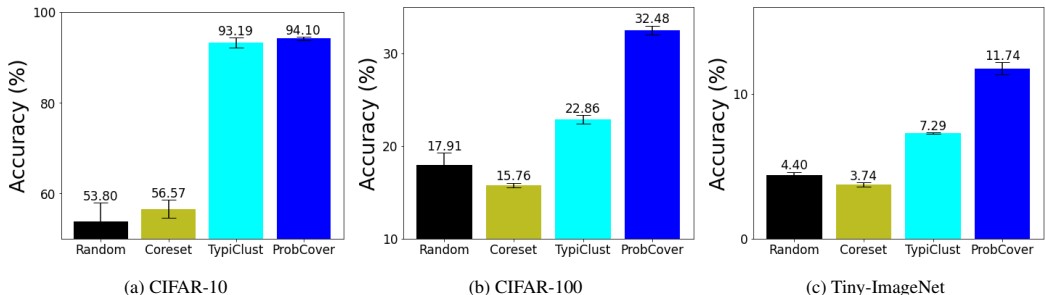

Figure 6: Framework (iii) Semi-supervised: comparison of AL strategies in a semi-supervised task. Each bar shows the mean test accuracy after 3 repetitions of FlexMatch trained using $b$ labeled examples, where $b$ is equal to the number of classes in each task. Error bars denote the standard error.

**Random initial selection** When following the uncertainty sampling principle, as many AL methods do, a trained learner is needed. Any such method requires therefore a non-empty initial pool of labeled examples to train a rudimentary learner, from which uncertainty selection can be bootstrapped. In the set of methods evaluated here (see Section 4.1), only two - *ProbCover* and *TypiClust* - are not affected by this problem. This can be seen in Fig. 4, noting that only these two methods do better than random in the initial step. Is this the only reason they outperform other methods in low budgets?

To address this question, we repeat the experiments reported in Fig. 4a-4b, using an initial random set of annotated examples across the board and by all methods. Results are reported in Fig. 7. When comparing Fig. 4a-4b and Fig. 7, we see that the advantage of *ProbCover* and *TypiClust* goes beyond the initial set selection, and remains in effect even if this factor is eliminated.

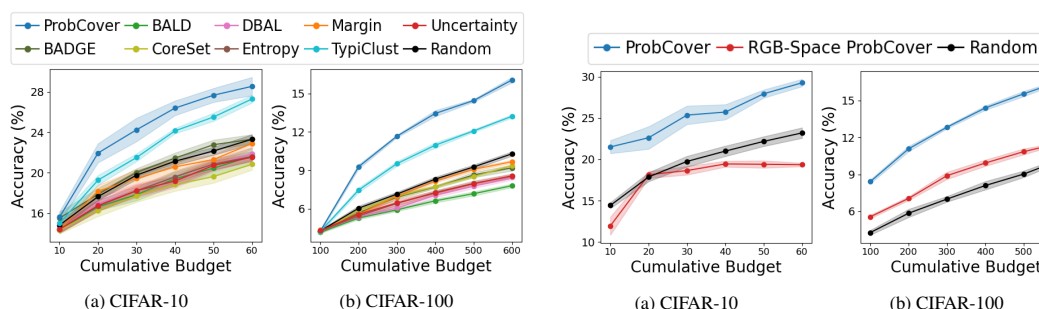

Figure 7: Random Initial pool in the supervised framework, an average of 1 sample per class.

Figure 8: Comparison of *ProbCover* when applied to the raw data vs the embedding space.

**RGB space distances** As discussed in Section 3, our approach relies on the existence of a good embedding space, where distance is correlated with semantic similarity. We now verify this claim by repeating the basic fully-supervised experiments (Fig. 4) with one difference: *ProbCover* can only use the original RGB space representation to compute distances. Results are shown in Fig. 8. When comparing the original *ProbCover* with its variant using RGB space, a significant drop in performance is seen as expected, demonstrating the importance of the semantic embedding space.

**The interaction between $\delta$ and budget size** To understand the interaction between the hyperparameter $\delta$ and budget $b$, we repeat our basic experiments (Fig. 4) with different choices of $\delta$ and $b$ using CIFAR-10. For each pair $(\delta, b)$, we select an initial pool of $b$ examples using *ProbCover* with $\delta$ balls, and report the difference in accuracy from the selection of $b$ random points. Average results across 3 repetitions are shown in Fig. 9 as a function of $b$. We see that as the budget $b$ increases, smaller $\delta$'s are preferred.

***Coreset* vs. *ProbCover*.** In Section 2.4 we argue that *ProbCover* is suitable for low budgets, while *Coreset* is suitable for high budgets. To verify this claim, we compare their performance under the following 3 setups while using the same embedding space, and report results on CIFAR-10:

- Low budget - Select an initial pool of 100 samples using the SimCLR representation.

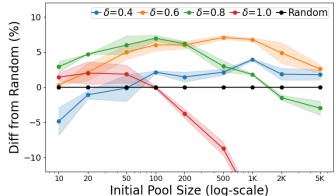

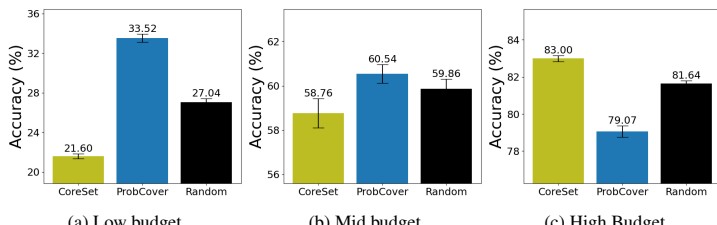

Figure 9: The accuracy difference between *ProbCover* when using different $\delta$ values, and the outcome of $b$ random samples (average over 3 repetitions).

(a) Low budget     (b) Mid budget     (c) High Budget

Figure 10: Comparing the performance under the supervised framework of *ProbCover* and *Coreset* on different budget regimes. The low budget shows an initial pool selection of 100 samples. Mid/High budget start with 1K/5K samples and query additional 1K/5K samples (see text).

- High budget - Train a model on 5K randomly selected examples. Then select an additional set of 5K examples using the learner's latent representation. This is the setup used by Sener and Savarese [37].

- Mid budget - Same as high budget, except the initial pool size and added budget are 1K.

Results are reported in Fig. 10. In the low budget regime, *ProbCover* outperforms *Coreset* as would be expected. In the mid-budget regime, where the feature space of the learner is informative, only *ProbCover* achieves significant improvement over random selection. In the high budget regime, *Coreset* improves over random selection, while *ProbCover* is least effective.

## 5   Summary and Discussion

We study the problem of AL in the low-budget regime. We model the problem as *Max Probability Cover*, showing that under certain assumptions on the data distribution, which are likely to hold in self-supervised embedding spaces, it optimizes an upper-bound on the generalization error of a 1-NN classifier. We devise an AL strategy termed *ProbCover*, which approximates the optimal solution. We empirically evaluate it in supervised and semi-supervised frameworks across different datasets, showing that *ProbCover* significantly outperforms other methods in the low-budget regime.

In future work we intend to investigate: (i) possible avenues for improving the choice of $\delta$ by making use of already known labels, or inferring a score for $\delta$ through the topology of the resulting covering graph; (ii) extensions of the current formulation of *Max Probability Cover*, by making $\delta$ - the radius of the balls - dependent on the samples rather than uniform; (iii) soft-coverage approaches, where the covering notion is not binary but some continuous measure, which may allow us to do away with $\delta$.

## Acknowledgments

This work was supported by the Israeli Ministry of Science and Technology, and by the Gatsby Charitable Foundations. We are grateful to our dedicated NeurIPS AC, who acted upon the lengthy discussion between us and the reviewers, as seen on OpenReview.

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
