## Appendix

## A  Some Definitions

In the proof of the main theorem below we use *Max Coverage*, a known NP-hard problem. We recapitulate its definition as follows:

**Definition A.1** (*Max Coverage*). Let $b \in \mathbb{N}$ denote an integer, $U$ denote a set of elements, and let $S = \{S_1, \dots, S_m\}$ denote a collection of subsets of $U$. In the problem of *Max Coverage* we wish to find $b$ subsets in $S$ with union of maximum cardinality

$$\operatorname*{argmax}_{S' \subseteq S; |S'| = b} \left| \bigcup_{S_i \in S'} S_i \right|$$

For convenience, we also repeat the definition of *Max Probability Cover* from Section 2.2:

**Definition A.2** (*Max Probability Cover*). Let $\mathbb{X}$ denote the input probability space. Let $X = \{x_i\}_{i=1}^m$, $x_i \in \mathbb{X}$ denote a set of points. Let $b \in \mathbb{N}$ denote an integer, and fix $\delta > 0$. In the problem of *Max Probability Cover* we wish to find a subset $L \subset X$, $|L| = b$, that maximizes the following probability

$$\operatorname*{argmax}_{L \subseteq X; |L| = b} P\left( \bigcup_{x \in L} B_\delta(x) \right)$$

## B  *Max Probability Cover* is NP-Hard

We first describe a constructive procedure to generate a collection of balls in $\mathbb{R}^m$ with a property we call *exhaustive intersection* (Def. B.2). We then use this collection in order to construct a reduction from *Max Coverage* to *Max Probability Cover*.

### B.1  *Exhaustive Intersection*: Constructive Procedure

Let $\{e_i\}_{i=1}^m$ denote the natural basis of $\mathbb{R}^m$. Let $B_r(p)$ denote the open ball of radius $r$ around $p$, and Let $B_r[p]$ similarly denote the closed ball.

**Definition B.1** (Inversion mapping). We define the *inversion* mapping $\iota : \mathbb{R}^m \setminus \{0\} \to \mathbb{R}^m \setminus \{0\}$ as

$$\iota(p) = \frac{p}{\|p\|_2^2}$$

**Lemma 2.** Let $X_i = \{x \subseteq \mathbb{R}^m \mid x \cdot e_i > 1\}$ denote a halfspace. Then $\iota(X_i) \subseteq B_{\frac{1}{2}}(\frac{1}{2}e_i)$.

*Proof.* Let $x \in X_i$. Membership in the two sets is defined by satisfying the equations $x \cdot e_i > 1$ and $d(\iota(x), \frac{1}{2}e_i) < \frac{1}{2}$. We will show that they are equivalent (see visualization in Fig. 11a). We use the polarization identity $a \cdot b = \frac{1}{2}(\|a\|^2 + \|b\|^2 - d(a,b)^2)$

$$1 < x \cdot e_i$$

$$\frac{1}{2\|x\|^2} < \left( \frac{x}{\|x\|^2} \right) \cdot \left( \frac{1}{2}e_i \right)$$

$$\frac{1}{2\|x\|^2} < \frac{1}{2}\left( \frac{1}{\|x\|^2} + \frac{1}{4} - d\left( \frac{x}{\|x\|^2}, \frac{1}{2}e_i \right)^2 \right)$$

$$d\left( \frac{x}{\|x\|^2}, \frac{1}{2}e_i \right)^2 < \frac{1}{4}$$

$$d\left( \iota(x), \frac{1}{2}e_i \right) < \frac{1}{2}$$

$\square$

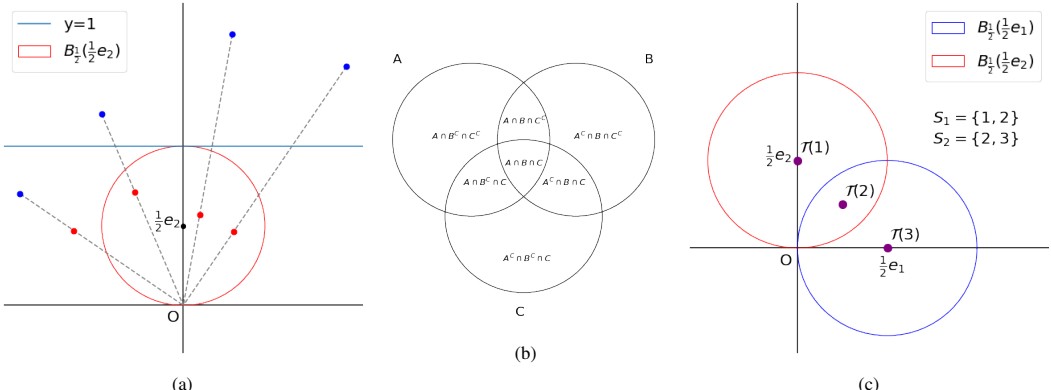

Figure 11: (a) Visualization of Lemma 2. The blue points above the plane $y = 1$ are mapped by $\iota(\cdot)$ to the red points inside $B_{\frac{1}{2}}(\frac{1}{2}e_2)$. Points below $y = 1$ are mapped outside the ball. (b) The exhaustive intersection in $\mathbb{R}^2$. (c) Visualization of the induced distribution for $m = 2$. Points $1, 3$ are mapped to disjoint parts of the two balls, while point 2 is mapped to their intersection $B_{\frac{1}{2}}(\frac{1}{2}e_1) \cap B_{\frac{1}{2}}(\frac{1}{2}e_2)$. Each point is then assigned a Dirac measure and the distribution is the normalized sum, as a result of which we get that $P(B_{\frac{1}{2}}(\frac{1}{2}e_1)) = P(B_{\frac{1}{2}}(\frac{1}{2}e_2)) = \frac{2}{3}$.

**Corollary 2.** $x \neq 0$, $x \cdot e_i < 1 \iff \iota(x) \notin B_{\frac{1}{2}}[\frac{1}{2}e_i]$.

**Definition B.2** (Exhaustive intersection). A collection of sets $(A_1, \ldots, A_m)$ is said to have the *exhaustive intersection* property if for any subset of indices $I \subset [m]$ there exists a point $x_I$ with

1. $x_I \in \bigcap_{i \in I} A_i$

2. $x_I \notin \bigcup_{j \in [m] \setminus I} A_j$

Put differently, $x_I \in A_i \iff i \in I$ (see the Venn diagram example in Fig. 11b).

**Lemma 3.** The collection of balls $\{B_{\frac{1}{2}}(\frac{1}{2}e_i)\}_{i=1}^m$ in $\mathbb{R}^m$ satisfies the *exhaustive intersection* property.

*Proof.* Let $I \subseteq [m]$ denote a subset of indices and define $\tilde{x}_I = \sum_{j \in I} 2e_j$, $x_I = \iota(\tilde{x}_I)$. From Lemma 2, for any $j \in I$, $\tilde{x}_I \cdot e_j = 2 \Rightarrow x_I \in B_{\frac{1}{2}}(\frac{1}{2}e_j)$, which proves the first part of Def. B.2. From Cor. 2, any $j \notin I$, $\tilde{x}_I \cdot e_j = 0 \Rightarrow x_I \notin B_{\frac{1}{2}}(\frac{1}{2}e_j)$, which proves the second part. $\qquad\square$

### B.2 Reduction from *Max Coverage* to *Max Probability Cover*

To improve readability, we restate in App. A the precise definitions of *Max Probability Cover* (Def. A.2) and *Max Coverage* (Def. A.1). Before we start, we note that in order for *Max Probability Cover* to be computable and not be trivially polynomial, we must assume that the encoding of the input distribution to *Max Probability Cover* is of polynomial size in the number of points.

**Theorem 2.** *Max Probability Cover* is NP-hard.

*Proof.* We construct a polynomial-time reduction from any *Max Coverage* problem $\mathbb{P}$ to another problem $\tilde{\mathbb{P}}$, which is an instance of *Max Probability Cover*. Let $In = (S_1, \ldots, S_m, b)$ be the input to $\mathbb{P}$, and let $U = \bigcup_{i=1}^m S_i$.

We define a mapping from the input of $\mathbb{P}$ to the input of $\tilde{\mathbb{P}}$: First, we fix the input space of $\tilde{\mathbb{P}}$ to $\mathbb{R}^m$, where $m$ is the number of sets in $\mathbb{P}$, and fix the set of $m$ points $X$ in $\tilde{\mathbb{P}}$ to $\{\frac{1}{2}e_i\}_{i=1}^m \subseteq \mathbb{R}^m$. We let $b$ remain the same, and fix the radius $\delta = \frac{1}{2}$. We abbreviate $B_i = B_{\frac{1}{2}}(\frac{1}{2}e_i)$. Next, we define a mapping $\mathcal{T} : U \to \mathbb{R}^m$ as $\mathcal{T}(u) = \iota(\sum_{i=1}^m 2 \cdot \mathbb{1}_{u \in S_i} e_i)$. From the proof of Lemma 3 we conclude that

$$\mathcal{T}(u) \in B_i \iff u \in S_i$$

Finally, we define the probability distribution $P$ as $P(A) = \frac{1}{|U|} \sum_{u \in U} \mathbb{1}_{\mathcal{T}(u) \in A}$ (see visualization in Fig. 11c).

**Lemma 4.** $P$ is a valid probability distribution.

*Proof.* $P$ is a finite sum of Dirac measures $\mathbb{1}_{\mathcal{T}(u)\in A}$, and as such it is a measure itself. Hence we only need to show that it is normalized, ie $P(\mathbb{R}^m) = 1$:

$$P(\mathbb{R}^m) = \frac{1}{|U|} \sum_{u\in U} \mathbb{1}_{\mathcal{T}(u)\in\mathbb{R}^m} = \frac{1}{|U|} \sum_{u\in U} 1 = 1$$

$\square$

In summary, the input of $\tilde{\mathbb{P}}$ is the following:

- Dataset: $X(In) = \{\frac{1}{2}e_i\}_{i=1}^m \subseteq \mathbb{R}^m$.
- Budget: $b(In) = b$.
- Ball radius: $\delta(In) = \frac{1}{2}$.
- Distribution: $(P(In))(A) = \frac{1}{|U|}\sum_{u\in U}\mathbb{1}_{\mathcal{T}(u)\in A}$

Before we continue, we require another short lemma:

**Lemma 5.** For all $u \in U$, $I \subseteq [m]$, $\mathbb{1}_{\mathcal{T}(u)\in\bigcup_{i\in I} B_i} = \mathbb{1}_{u\in\bigcup_{i\in I} S_i}$

*Proof.* We prove the conditions are equivalent:

$$\mathcal{T}(u) \in \bigcup_{i\in I} B_i \iff \exists i \in I \quad \mathcal{T}(u) \in B_i \iff \exists i \in I \quad u \in S_i \iff u \in \bigcup_{i\in I} S_i$$

$\square$

To show that a reduction is valid for an optimization problem, we need to show that the objectives are equivalent. The objective in *Max Coverage* is the size of the union, whereas in *Max Probability Cover* it is the probability of the $\delta$-ball union.

$$P(\bigcup_{i\in I} B_i) = \frac{1}{|U|} \sum_{u\in U} \mathbb{1}_{\mathcal{T}(u)\in\bigcup_{i\in I} B_i}$$

$$= \frac{1}{|U|} \sum_{u\in U} \mathbb{1}_{u\in\bigcup_{i\in I} S_i}$$

$$= \frac{1}{|U|} \left| \bigcup_{i\in I} S_i \right|$$

Since $|U|$ is constant the optimization is equivalent.

Finally, we show that the induced probability can be specified in polynomial space: as $|U|$ is polynomial in the size of the input, it follows that the distribution as a normalized sum of indicator functions can be specified as a table of the embedding of size $|U| \cdot m$, which is polynomial. $\square$

## C   Implementation Details

Source code used in this work is available at the following url:

https://github.com/avihu111/TypiClust

### C.1   Selection Method

**Representation Learning: CIFAR10, CIFAR100, TinyImageNet.** To extract semantically meaningful features, we trained SimCLR using the code provided by Van Gansbeke et al. [41] for CIFAR-10, CIFAR-100 and TinyImageNet. Specifically, we used ResNet-18 [20] with an MLP projection layer to a 128-dim vector, trained for 500 epochs. All the training hyper-parameters were identical to those

used by SCAN (all details can be found in Van Gansbeke et al. [41]). After training, we used the 512 dimensional penultimate layer as the representation space.

**Representation Learning: ImageNet.** We extracted features from the official (ViT-S/16) DINO weights pre-trained on ImageNet. We used the L2 normalized penultimate layer for the embedding. All the exact hyper-parameters can be found at Caron et al. [5].

**Randomness in *ProbCover* Selections.** In order to reduce the correlation between different repetitions using *ProbCover*, we added the following modification to the selection algorithm: instead of taking the node with the highest degree at each iteration, we selected randomly one of the 5 nodes with the highest degree. We verified that both algorithms achieved similar performance, where the deterministic version has slightly better results.

## C.2 Fully Supervised Evaluation

We trained a ResNet18 on the labeled set, using the AL comparison framework created by Munjal et al. [33], and following the protocol described in [19] (see details in [19] and the shared code).

## C.3 1-NN Classification with Self-Supervised Embeddings

In these experiments, we also used the framework by Munjal et al. [33]. We extracted an embedding similar to § C.1, with which we trained a 1-NN classifier using the default parameters of *scikit-learn*.

## C.4 Semi-Supervised Classification

When training FlexMatch [52], we used the AL framework by Zhang et al. [51]. All experiments involved 3 repetitions.

**CIFAR-10.** We used the standard hyper-parameters used by FlexMatch [52]. Specifically, we trained WideResNet-28 for 400k iterations using the SGD optimizer, with $0.03$ learning rate, $64$ batch size, $0.9$ momentum, $0.0005$ weight decay, $2$ widen factor, and $0.1$ leaky slope. The weak augmentations used are identical to those used in FlexMatch and include random crops and horizontal flips, while the strong augmentations were generated by RandAugment [11].

**CIFAR-100.** Similar to CIFAR-10, but increasing the widen factor to $8$.

**TinyImageNet.** We trained ResNet-50, for 1.1m iterations. We used an SGD optimizer, with a $0.03$ learning rate, $32$ batch size, $0.9$ momentum, $0.0003$ weight decay, and $0.1$ leaky slope. The augmentations were similar to those used in FlexMatch.

# D Additional Empirical Results

## D.1 Improving the greedy approximation

The greedy approximation used in *ProbCover* guarantees $1 - \frac{1}{e}$ approximation to the maximum cover problem. Hunt et al. [24] showed that a polynomial time approximation scheme (PTAS) exists for this problem, suggesting the possibility of better polynomial approximations. However, Marx [31] proved that there is no efficient PTAS to this problem, implying that such polynomial approximations may not be practical. For example, to achieve a $1 - \frac{1}{e}$ approximation using the PTAS suggested in Marx [31] would require $O(n^{100})$ time. Thus, a significantly better approximation than the greedy solution is left for future work. Instead, we improved the greedy algorithm by choosing at each iteration the optimal 2 balls in a greedy way. While this greedy solution achieves a better approximation in theory, in practice we did not see any improvement over the single-ball greedy solution.

## D.2 ImageNet subsets

When evaluating *ProbCover* on ImageNet-50 and ImageNet-100, we report a similar qualitative behavior as seen in other datasets: *ProbCover* performs better than all baselines in the very low-budget regime, using 5 AL rounds with a budget equal to $b = 50$ examples. More concretely, in Fig. 12 we show results corresponding to Figs. 3-5 when using ImageNet-50.

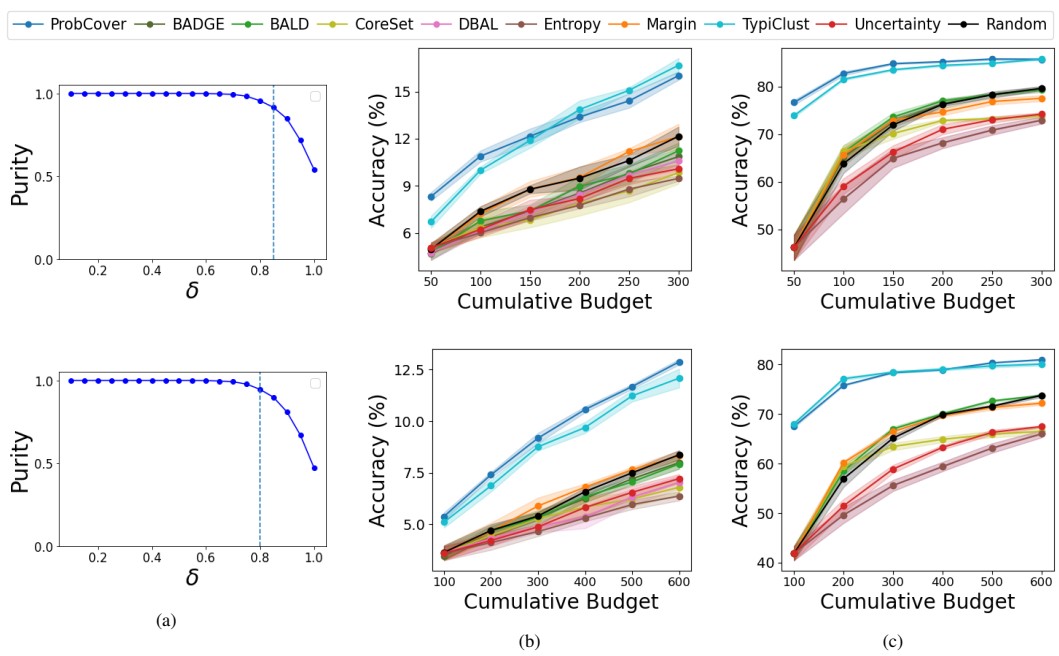

Figure 12: A Comparative evaluation of *ProbCover* on ImageNet-50 (top row) and ImageNet-100 (bottom row). (a) Similar to Fig 3, in which we estimate $\delta$. (b) Similar to Fig. 4, trained in the fully supervised framework. (c) Similar to Fig. 5, trained in the semi-supervised by transfer learning framework.

### D.3 *TypiClust* vs *ProbCover* on SCAN feature space

Both *ProbCover* and *TypiClust* use an unsupervised self-representation embedding as part of an active learning query selection algorithm. In Section 4.2, when comparing *ProbCover* to *TypiClust*, we used the same embedding in both of them, to avoid possible confounds relating to the choice of the specific representation algorithm.

As *TypiClust* reached the best performance using SimCLR representation in most budgets and frameworks on CIFAR-10 and CIFAR-100, we chose that embedding space to compare to *ProbCover*. However, in the fully-supervised framework, with a budget of 10 examples, *TypiClust* yields better results using the embedding space of SCAN.

In Fig. 13, we plot a comparison between *ProbCover* and *TypiClust* in this budget, when both are using the embedding space of SCAN. We find a similar trend to the results reported in Section 4.2: *ProbCover* achieves higher accuracy than *TypiClust*, and both surpass random sampling in this budget.

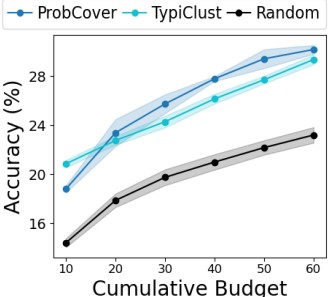

Figure 13: Comparison of *ProbCover* and *TypiClust* using the SCAN feature space. ($\delta = 0.8$)

**D.4 Comparison with W-Dist**

In Section 4, we compare *ProbCover* to several other active learning baselines, including W-Dist [30]. As this method is computationally demanding, we are only able to evaluate its performance on the CIFAR-10 and CIFAR-100 datasets, which are the smallest datasets we consider.

We note that the results differ from the results reported in the original paper. This stems from several things: firstly, we use 1-NN classification instead of linear classification in the self-supervised scenario. Secondly, the implementation of the Wasserstein method is ours, based on the pseudo-code published in the original work, as no official implementation is available, though we did our best to follow the instructions of the original paper. Thirdly, the method is unique in that it requires a long time to select samples (the original version set a 3 hours timeout for the selection of every 10-20 samples). Instead of the long timeouts suggested in the original work, we used 20 minutes timeouts per round, which reached similar results.

# E  Time and Space complexity of *ProbCover*

During the training of a neural network, *ProbCover* is executed a single time in order to select the best subset to query for human annotation for subsequent network training.

For the complexity calculation below, let $n$ denote the number of examples in the unlabeled and labeled pool $|U \cup L|$, $d$ the dimension of the data embedding space, and $b$ the query budget. *ProbCover* can be split into two steps:

## E.1  Adjacency graph

Constructing the adjacency graph requires computing pairwise distances in the embedding space.

**Time Complexity**: $O(n^2 d)$ time. In practice, it takes roughly 10 minutes on a single NVIDIA A4000 GPU even on the largest dataset we consider – ImageNet with DINO embedding, where $n = 1281167, d = 384$.

**Space Complexity**: naively we have $O(n^2)$, which is impractical for large datasets like ImageNet. However, we only need to save edges whose distance is smaller than $\delta$. We store the edges using a sparse matrix in coordinate list (COO) format, so the space complexity is $O(|E|)$, where $E$ is the set of edges in the graph.

Although $O(|E|)$ is still $O(n^2)$ in the worst case, in practice, the average degree of each vertex in the graph using radius $\delta$ is a few orders of magnitude smaller than $n$, resulting in manageable space complexity. For example, When selecting samples from ImageNet with $\delta = 0.55$, the average degree was 24 and the algorithm total memory consumption was $12GB$.

## E.2  Sample selection

We iteratively select samples from the current sparse graph, removing incoming edges to newly covered samples. We note that unlike the adjacency graph creation, the sample selection cannot be parallelized, as each selection step depends on the previous step.

**Time Complexity**

Breaking down the steps in the selection of a single sample:

- Calculating node degrees – $O(|E|)$ time.
- Finding node with a maximal degree – $O(n)$ time.
- Removing covered points' incoming edges from the graph – $O(|E|)$ time.

All in all, the complexity is $O(|E| + n)$ for selecting a sample, and $O(n^2 k)$ overall in the worst case. As we select more and more points, more edges are removed, making the selection of later samples faster. In practice, thanks to the vectorization of these steps it takes roughly 15 minutes to select $k = 1000$ samples from ImageNet on a single CPU, and a couple of seconds in CIFAR-10/100.