# OpenReview forum: "Active Learning Through a Covering Lens"
_NeurIPS.cc/2022/Conference — NeurIPS 2022 Accept_

### Official Review · Reviewer_G2iF · 2022-07-10

**Rating:** 6
**Confidence:** 4
**Soundness:** 3 good
**Presentation:** 3 good
**Contribution:** 3 good

**Summary:**

The authors propose Max Probability Cover which is the optimal strategy for 1-NN model in that it minimizes an upper bound on their generalization error.
As a data distribution is not known in practice, they use empirical data distribution which converts Max Probability Cover to Max Coverage.
To work around NP-hard nature of Max Coverage, they propose a greedy algorithm for the Max Coverage named ProbCover.
For 1-NN model, Max Probability Cover and Coreset are in “dual” relationship in that Max Probability Cover seek to maximize the coverage for a fixed \delta whereas Coreset seek to minimize \delta for the fixed coverage to 1.
In the low budget regime where the budget is not enough to cover the unlabeled set, the proposed ProbCover is effective than the existing AL methods most of which are designed for high budget regime.
The authors demonstrate it using 3 different frameworks compared to 9 different baseline AL models.


**Questions:**

1. As discussed in Sec 3, the authors mentioned that their approach relies on the existence of a good embedding space. In Sec 4.3, although the authors demonstrated that being in a bad embedding space indeed makes the performance worse, I am not convinced that ResNet-18 trained on few hundreds of images would provide good representations for high dimensional inputs like images.
It seems like being in the low budget regime and being in a good embedding space are controversial for over-parameterized neural networks. So, I think either the statement in Sec 3 is wrong or somehow ResNet-18 is in a good embedding space, which I don’t think is clearly demonstrated or justified. Could the authors elaborate it?

2. The authors clearly specify that the proposed method is for low budget regime. However, in practice, the budget $b$ would increase as time goes and so does the covered region. Then, there must be a point where a given problem transits from low budget regime to high budget regime. In that case, what do the authors think would be an efficient way (measurement) to detect the transit?

**Limitations:**

Limitations have been mentioned throughout the paper.

**Strengths And Weaknesses:**

Strength
1. This works is one of the first few works addressing the low budget regime problem in AL.
2. It theoretically demonstrates the close relationship (loose duality) between Max Probability Cover and Coreset in 1-NN model case.
3. Hyperparameter tuning step is often ignored in AL but the authors address it (in this case \delta) in a reasonable way.


Weakness
1. The theoretical analysis is based on 1-NN model, which is not strong enough in practice. Also, there is a gap between 1-NN model and deep neural networks so there is a limitation of extending the theoretical results in Sec 2 to general architectures.

---

> ### Author Response · Authors · 2022-08-02
> **Response to Reviewer G2iF**
>
> **A good embedding space**
>
>  As you mentioned, the embedding space of ResNet-18 trained on a few hundred examples (or even less than a hundred in many cases) does not provide a good representation for high-dimensional inputs like images. Instead, ProbCover uses a pre-computed embedding space, which is provided by an unsupervised task (SimCLR/DINO/SCAN) trained on the entire unlabeled dataset, which is presumably much larger than hundreds of examples.
>
> This kind of representation has been shown by us and by previous art to have "semantic" properties, and in general, is much better than fully-supervised networks trained on a few hundreds of examples.
>
> We note that even between active learning rounds, we still use the *unsupervised* embedding space to query examples for human annotation, as the embedding space of a network trained on a few samples is highly biased.
>
> While this point appeared in the original manuscript, it was not sufficiently clear -- as similar concerns appeared in the review of Reviewer eB6a. To clarify this point, we made several changes to the manuscript. The most relevant changes can be found on lines 165-168, 175-178, and in a re-write of Algorithm 1 -- to include this point in a direct matter.
>
> Additionally, to make the implementation details clearer, we added all the (anonymized) code of our method in the supplemental material of the current revision.
>
> **Transition from low to high budget**
>
> Your observation is correct. Given some labeled data and unlabeled pool, it is unclear how to determine the transition point: whether the current data is in the low budget regime or the high budget regime. It is something that we wish to characterize within our framework, and are currently working on for future work, with some promising results. As this future work is both out of scope for this manuscript, and not yet peer-reviewed, but will be subject to a double-blind review process in the future, we should not discuss it in detail in this public forum.

---

> > ### Comment · Reviewer_G2iF · 2022-08-05
> > **Followup Questions**
> >
> > Thank you for clarifying the questions I had.
> > But it is still unclear how the features are learned in active learning process.
> >
> > The authors said "We note that even between active learning rounds, we still use the unsupervised embedding space to query examples for human annotation, as the embedding space of a network trained on a few samples is highly biased."
> > It seems like the network is not re-trained on labeled data points due to bias.
> > Does this mean a model is not trained on the labeled dataset at all in the process of active learning? Or does it mean that there are two separate models: one trained in unsupervised learning (no changes in features), and the other one trained on the labeled set (+ unlabeled set for semi-supervised learning case) and used for evaluation?
> > In algorithm 1, X represents embedding of representation learning algorithm on $U \cup L$. I thought this is the feature from unsupervised learning but as you said "as the embedding space of a network trained on a few samples is highly biased", I am confused now.
> > Could you elaborate it?
> >
> > I have one additional question. I may have missed it but do you have a proof that the objective is a submodular monotone function?

---

> > > ### Author Response · Authors · 2022-08-05
> > > **Answers to followup questions**
> > >
> > > Thanks for your willingness to work with us in order to improve and clarify the paper.
> > >
> > > Just as a notation reminder, we use $U$ to denote a large set of unlabeled data, $L$ to denote some much smaller set of labeled data (possibly empty), and $b$ to denote the query budget $b\in\mathbb{N}$.
> > >
> > > In answer to your question: Essentially, there is one (classification) model which is trained to classify the data, this is our target and an additional auxiliary model which is used to select queries.
> > >
> > > 1) In our proposed method, the first thing we do is to train the auxiliary model to learn an unsupervised representation learning task, using all the points in $U\cup L$. We then use this model's penultimate layer as feature space for both $U$ and $L$ and keep it fixed. We do not update this model even as additional labels become available.
> > >
> > > 2) The second thing we do is to run the greedy max coverage algorithm on $U\cup L$: we draw $\delta$-balls around each example in $L$ and find the $b$ examples from $U$ that would cover as many points as possible from $U\cup L$, in the feature space provided by the auxiliary model.
> > >
> > > 3) The third thing we do is to add these $b$ examples to $L$ and train the main classification model, **from scratch**, on $L$ in the fully-supervised experiments, or $U\cup L$ in the semi-supervised experiments. This classification model is never used as a feature space, as it may be highly biased (it was only trained on a very small set $L$).
> > >
> > > In each subsequent additional active learning iteration, when we need to choose another set of $b$ query examples, we simply reiterate steps two and three, using the larger set of labeled examples $L$ from the previous iteration.
> > >
> > > If this is still confusing, please let us know and we will gladly provide additional details. In addition, if you find that the revised paper is still confusing on any of these points, it would be great to know so we can expand the explanation and clarify it for future readers.
> > >
> > > **submodular function**
> > >
> > > The objective of Max Coverage is submodular monotone function. This intuitively makes sense: the more balls you can use, the more examples you can cover (hence monotone), and the balls you cover with are going to cover fewer and fewer examples each time, as the denser areas are covered early on (hence submodular).
> > >
> > > This was proved formally in prior work by Krause and Golovin (2014) [1], under section 1.1 in their work. This proof is referenced on lines 169-173 in our current revision.
> > >
> > >
> > > [1] Andreas Krause and Daniel Golovin. Submodular function maximization. Tractability, 3: 71–104, 2014.

---

> > > > ### Comment · Reviewer_G2iF · 2022-08-06
> > > > **Followup Questions 2**
> > > >
> > > > Thank you for the clarification.
> > > >
> > > > I do think this sort of description should be in Section 3. As the Reviewer eB6a also mentioned, due to the absence of training details, it was hard to fully understand the details such as having two separate models: the main (classification) model and auxiliary model.  Although the authors responded, the details are mentioned in Appendix D, the information is all scattered in that section, which makes it still hard to capture. Also, line 222-230 do not really cover the details.
> > > >
> > > > Now that I know that the main and auxiliary models are maintained separately, I am worried about the training process. I guess here the assumption is that the feature space of the auxiliary model is a good proxy for that of the main model. But, how is it guaranteed? Unless there is only one optimal feature mapping so that the feature mapping of both auxiliary and main models converges to that optimal feature mapping, they (feature mapping) can be arbitrary different. Even if the feature space of the auxiliary model is a good proxy at one (query) point, it can deviate quite a bit from that of the main model as the number of labeled data points are added. Could you elaborate this?

---

> > > > > ### Author Response · Authors · 2022-08-07
> > > > > **Answers to followup questions 2**
> > > > >
> > > > > **Training details**
> > > > >
> > > > > In active learning, the concept of a representation space is commonly used. In the high-budget regime, as the classification model is already "strong enough", this representation space is often chosen to be the penultimate layer of the classification model (e.g., [1-2] in the list below).
> > > > >
> > > > > However, in the low-budget regime, such representation could be sub-optimal, as the classification model is biased, due to the small and unrepresentative labeled set. Therefore, in **all** the low-budget works we are aware of (e.g., [3-6] in the list below), the representation of an auxiliary model is used. This representation space is often achieved by training the auxiliary model on an unsupervised task on the unlabeled dataset or via transfer learning.
> > > > >
> > > > > By way of background: the use of 2 models, one for representation and one for the actual task (whether classification or otherwise), is a very common practice in deep learning framework in general, with early examples that goes back to 2014 (e.g., [7] below). In fact, it is so commonly used both in low-budget active learning and in deep learning in general that we did not originally fully appreciate the need for a detailed explanation, believing that a brief explanation would be sufficient. However, and although it is not the focus of the paper, following this discussion, we will expand somewhat the concise description in section 3 regarding the role of the representation space, hopefully making it easier to capture.
> > > > >
> > > > > Please note that as many of the prior work on low-budget active learning uses similar unsupervised representations, this is not a main contribution of our work. Our contribution is in steps 2) and 3) of our previous answer: given some representation, ProbCover performs a greedy selection of points that best cover $U\cup L$. These points are later used to train a classification model. This **is** the ProbCover method, and its description in Section 3 is thorough and complete.
> > > > >
> > > > > **Feature space**
> > > > >
> > > > > The problem you raise is a fundamental disadvantage in all low-budget active learning approaches, and more broadly on any approach which relies on an unsupervised feature space or one provided by transfer learning. This includes, among other things, all transfer-learning-based approaches. Indeed, when using a feature space of an auxiliary model, there is no guarantee that the features will be a good proxy for the model at hand. While there is no such guarantee, many such approaches work fairly well in practice.
> > > > >
> > > > > The only remaining way to provide reassurance is via a thorough empirical study, which we believe we have provided. We showed that ProbCover beats **all** other alternative AL methods in the low-budget regime by a large margin. Importantly, when comparing with methods that use an unsupervised feature space, we evaluated all methods using identical feature space. Our empirical results are consistent across several benchmarks, which are the standard test-bed for active learning methods.
> > > > >
> > > > > We note that ProbCover only requires that the representation space is a "good" representation of the dataset, in the sense that similar examples are closer (under $L_2$ distance) than dissimilar examples. As long as this property holds, ProbCover will pick the most representative examples, which have been shown in previous art to be helpful for the low-budget regime [3-4].
> > > > >
> > > > > While we are not guaranteed that the representation is a good proxy to the current classification model, ProbCover is quite robust and works using many different representation spaces. In the experiments done in the paper, we see that several different auxiliary models and tasks work well: CIFAR-10/100 and TinyImageNet are trained using SimCLR [8], and ImageNet is trained using DINO [9]. In Fig.12 in the appendix, we report that several representation spaces are suited to the same dataset: ProbCover on CIFAR-10 reaches high results with the representation of SimCLR [8] and with the representation of SCAN [10].

---

> > > > > > ### Author Response · Authors · 2022-08-07
> > > > > > **Bibliography for "Answers to followup questions 2"**
> > > > > >
> > > > > > Bibliography:
> > > > > >
> > > > > > [1] Sener, Ozan, and Silvio Savarese. "Active learning for convolutional neural networks: A core-set approach." arXiv preprint arXiv:1708.00489 (2017).
> > > > > >
> > > > > > [2] Ash, Jordan T., et al. "Deep Batch Active Learning by Diverse, Uncertain Gradient Lower Bounds." International Conference on Learning Representations. 2019.
> > > > > >
> > > > > > [3] Mahmood, Rafid, Sanja Fidler, and Marc T. Law. "Low-Budget Active Learning via Wasserstein Distance: An Integer Programming Approach." International Conference on Learning Representations. 2021.
> > > > > >
> > > > > > [4] Guy Hacohen, Avihu Dekel, Daphna Weinshall. "Active Learning on a Budget: Opposite Strategies Suit High and Low Budgets." International conference on machine learning. PMLR, 2022.
> > > > > >
> > > > > > [5] Pourahmadi, Kossar, Parsa Nooralinejad, and Hamed Pirsiavash. "A Simple Baseline for Low-Budget Active Learning." arXiv preprint arXiv:2110.12033 (2021).
> > > > > >
> > > > > > [6] Rebuffi, Sylvestre-Alvise, et al. "Semi-supervised learning with scarce annotations." Proceedings of the IEEE/CVF Conference on Computer Vision and Pattern Recognition Workshops. 2020.
> > > > > >
> > > > > > [7] Sharif Razavian, Ali, et al. "CNN features off-the-shelf: an astounding baseline for recognition." Proceedings of the IEEE conference on computer vision and pattern recognition workshops. 2014.
> > > > > >
> > > > > > [8] Chen, Ting, et al. "A simple framework for contrastive learning of visual representations." International conference on machine learning. PMLR, 2020.
> > > > > >
> > > > > > [9] Caron, Mathilde, et al. "Emerging properties in self-supervised vision transformers." Proceedings of the IEEE/CVF International Conference on Computer Vision. 2021.
> > > > > >
> > > > > > [10] Van Gansbeke, Wouter, et al. "Scan: Learning to classify images without labels." European conference on computer vision.
> > > > > > Springer, Cham, 2020.

---

> > > > > > ### Comment · Reviewer_G2iF · 2022-08-07
> > > > > > **Response**
> > > > > >
> > > > > > Thank you for the detailed description.
> > > > > >
> > > > > > The discrepancy between two different models is a critical problem, which I did not capture in the first place.
> > > > > > Although it seems like it works well in practice as the authors pointed out above, as researchers, we want to understand why it actually works. Unfortunately, at this point, it is still unclear how and why a feature space from unsupervised learning without further training can be a good proxy for that of supervised learning with training on gradually increasing number of data points.
> > > > > > Thanks to the detailed explanation, I do understand this is not just a problem of this work but I initially gave a relatively high score as I did not realize this limitation. Hence, although it is still a good work, I decrease my score by 1.
> > > > > > It would be great if the discussion above is included in the paper to clarify the existing fundamental problem of low-budget active learning.

---

> > > > > > > ### Author Response · Authors · 2022-08-08
> > > > > > > **Response**
> > > > > > >
> > > > > > > First of all, although we may sound negative, we want to point out that we are very grateful for the time and effort you took for both reviewing our paper, and in engaging a meaningful conversation about it. However, we are terribly sorry to hear about the score reduction, which in our opinion is a bit harsh.
> > > > > > >
> > > > > > > In our view, the fact that representations learned by one model are beneficial to the training of another model is a very strong aspect of deep learning, rather than a limitation. This property, which allows the use of transfer learning, shows a greater aspect of generalization that deep models do: knowledge from one task is helpful for the performance of different tasks.
> > > > > > >
> > > > > > > The latest progression in unsupervised representation learning is what made a work like our own even possible: it introduced a way to extract knowledge from the dataset, even when no labels are available, allowing a kind of transfer knowledge from the unlabeled pool to the labeled pool.
> > > > > > >
> > > > > > > We agree that as researchers, our main goal is not just to make things work, but rather to understand how they work. As such, under the assumption that representation learning can extract knowledge from unlabeled data, we propose a method that can capture it, **and** provide a detailed theoretical ground work that aims to explain why such a method works.
> > > > > > >
> > > > > > > However, the question you are raising is much more fundamental -- you want to understand why models even learn representations that can extract knowledge from datasets to begin with. This is a fascinating (and very important, in our view) question, which we are very interested in. Yet one must keep in mind that this has been an open question in the deep learning field for a large part of the last decade, which goes much beyond low-budget active learning.
> > > > > > >
> > > > > > > We think that failing to answer such an important timely question should not be held against us as a limitation of this specific work, and would like to ask you to reconsider the downgrade of your score.

---

> > > > > > > > ### Comment · Reviewer_G2iF · 2022-08-09
> > > > > > > > **Response**
> > > > > > > >
> > > > > > > > I was pointing out the discrepancy between the main and auxiliary models not transferring learning or representation learning in general.
> > > > > > > >
> > > > > > > > > In our view, the fact that representations learned by one model are beneficial to the training of another model is a very strong aspect of deep learning, rather than a limitation.
> > > > > > > >
> > > > > > > > I cannot agree more on this point as a general transfer learning framework. But, what this work assumes for the main and auxiliary models is that the features learned by the auxiliary model won't be that different from the features learned by the main model. It is not a transfer of knowledge: the auxiliary model is a proxy to the main model. My question was how and why the auxiliary model can be a proxy to the main model.
> > > > > > > >
> > > > > > > > > However, the question you are raising is much more fundamental -- you want to understand why models even learn representations that can extract knowledge from datasets to begin with.
> > > > > > > >
> > > > > > > > Again, this is not what I have been asking.
> > > > > > > >
> > > > > > > > I understand that it is disappointing to have score reduction but the concern I am raising is not unreasonable. I still do think this work is a good work, but with its own limitation.

---

### Official Review · Reviewer_cppJ · 2022-07-12

**Rating:** 6
**Confidence:** 4
**Soundness:** 3 good
**Presentation:** 3 good
**Contribution:** 3 good

**Summary:**

This paper focuses on low-budget active learning tasks. The proposed method attempts to model the problem as Max Probability Cover question. As Max Coverage is known to be NP-hard problem, the author then adapts a greedy algorithm for subset selection for annotation through covering lens. The improvement of the proposed method over the baselines is shown in supervised and semi-supervised frameworks across different datasets

**Questions:**

Why Max Coverage's objective is a submodular monotone function? Need a reference here.

The current greedy algorithm can not guarantee the optimality of Max coverage, the author may explore or discuss a better way to achieve optimal.

**Limitations:**

Adequately addressed the limitations and potential negative societal impact.

**Strengths And Weaknesses:**

Strengths:

1. The idea of seeking maximize Probability Coverage for low budget AL is straightforward and the author gives theoretical proof for this.

2. It is interesting to treat Max Probability Cover and Coreset as dual problems.

3. Most of this paper is clear for practice and the insight for model design is well illustrated.

4. The improvement of the proposed method over the current state-of-the-art methods looks promising in low budget AL setting.

Weaknesses:

1. The greedy algorithm can not guarantee the optimality of Max coverage. Therefore, the sampling solution obtained by the greedy algorithm is hard to be optimal.

2. Missing reference and comparison with ICML'22 Low Budget Active Learning via Wasserstein Distance: An Integer Programming Approach

---

> ### Author Response · Authors · 2022-08-02
> **Response to Reviewer cppJ**
>
> **Suboptimality of the Greedy algorithm**
>
> Max coverage is NP-hard, hence finding an optimal solution is not feasible. In appendix D.3 of the original manuscript (App E.1 in the current revision), we discussed this problem in greater detail. To summarize, we reported that while there is a regular polynomial-time approximation scheme (PTAS) for Max Coverage, it was proven that no efficient PTAS (EPTAS) exists. Therefore, finding an $\varepsilon$-approximation for Max Coverage is also not feasible.
>
> To the best of our knowledge, an algorithm with a significantly better approximation than the greedy algorithm is yet to be found. Using the PTAS to find a better approximation than the greedy algorithms takes more than $O(n^{100})$ time. Instead, we performed several experiments with a marginally better greedy algorithm: instead of picking the best ball in each iteration, we pick the best combination of 2 balls in each iteration. While this is significantly more computationally demanding (and can only be run on small datasets), we did not see any difference in the solutions.
>
>
>
> **Missing reference: Wasserstein Distance**
>
> A reference to Mahmood et al. appears in the original manuscript of the paper. In this version, we added a comparison to Mahmood et al. in the experiments depicted in Figs. 3 and 4. We find that Mahmood et al. indeed outperforms random sampling in the low-budget regime, although under-performs ProbCover by a significant margin.
>
> We note that our reported results on Mahmood et al. are different than the original paper, as we did not have their code available (they didn't publish it, nor agreed to send it to us over mail). The code used in this paper is an implementation based on the pseudo-code present in the paper's appendices. As the algorithm is computationally demanding, we only ran it on CIFAR-10 and CIFAR-100. Note that ImageNet-based datasets were not included in the original paper.
>
> These changes appear on lines 213-215, Figs. 3,4 and in Appendix E.4.
>
> **ProbCover is Submodular Monotone**
>
> Max Coverage being a submodular monotone function is discussed in [26] (Submodular function maximization, Krause et al) under Section 1.1-"Weighted coverage functions", which is cited in the sentence where the claim is made (lines 168-170 of the paper).
>
> In lines 168-170 in the original paper, [26] is used to support two different claims in the same sentence. A clarification was added in the new revision for clarity (lines 170-171).

---

> > ### Comment · Reviewer_cppJ · 2022-08-09
> > **Response to authors**
> >
> > Thanks, most of my concerns have been addressed.  I will keep the original score.

---

### Official Review · Reviewer_eB6a · 2022-07-18

**Rating:** 4
**Confidence:** 3
**Soundness:** 3 good
**Presentation:** 3 good
**Contribution:** 2 fair

**Summary:**

This work introduces a new active learning algorithm for the low-budget regime. The main idea is to maximize probability coverage. The method is demonstrated in both low-budget and high-budget regimes. Moreover, the author also conducts experiments on fully/semi-supervised settings to demonstrate the effectiveness of the method.

**Questions:**

1. Please illustrate the data selection process with the proposed method. Using the feature of each data point or just the RGB image?  What are the points of the graph in the specific experiment?
2. The building process of the graph and data selection is time-consuming. Please elaborate on the cost of these two processes.
3. Please conduct experiments on a larger dataset or other tasks to demonstrate the method, especially for realistic applications. Like the full ImageNet or detection on the COCO dataset.
4. Comparing Fig.3(a) of this work with Fig.4(a) of TypiClust, the performance of TypiClust reported in this paper is lower than the result in TypiClust. Please clarify the difference between these two curves. Compared with the performance reported in TypiClust Fig.4(a), the improvement is marginal.

I would like to change the score if the concerns could be resolved in the rebuttal.

**Strengths And Weaknesses:**

Pros:
- The task is important and fundamental in the research community.
- The low-budget regime is challenging for the realistic scenario.
- The experiments seem extensive and well-designed.

Cons:
- The training details are missing and making it difficult to follow.
- The experiments are only conducted in a small dataset.
- The comparison with TypiClust is confusing.

---

> ### Author Response · Authors · 2022-08-02
> **Response to Reviewer eB6a**
>
> **RGB or feature space**
>
> ProbCover uses a feature space for sample selection, which is computed from the unlabeled data before the active selection of queries. Hence, unless specifically mentioned otherwise, in all our experiments each vertex in the graph represents a single image, and the edge between every pair of vertices appears only if the distance between them -- in the pre-computed feature space -- is smaller than some fixed $\delta$.
>
> Specifically, in the original manuscript - line 216, we list the feature spaces we used for different datasets. Additionally, in line 255 and Figure 7, where we report our ablation study, we compare the performance of using a deep embedding space and the original RGB input space.
>
> While this point appeared in the original manuscript, it was not sufficiently clear -- Reviewer G2iF had similar concerns. To clarify this point, we made several changes. The most relevant changes can be found on lines 165-168, 175-178, and in a re-write of Algorithm 1 -- to include this point in a direct matter.
>
> **Training details**
>
> All training details appear in the original manuscript, in App D. To make this information more accessible in the current revision, we clarified this reference in the main paper (lines 222 and 230). Additionally, we added an anonymized code to the supplementary material of the current submission, containing all the implementation details. All code will be made publicly available upon acceptance.
>
> **Time and Space complexity and scaling**
>
> Given a dataset with $n$ examples with an embedding space of dimension $d$, choosing $k$ examples according to ProbCover has time complexity of $O\left(n^2(d+k)\right)$ and space complexity of $O(n^2)$. While the time complexity is reasonable in most datasets, the space complexity of $O(n^2)$ may be impractical. This space complexity stems from the construction of the graph, which requires the calculation of distances between every pair of examples. As we only need to save edges between points whose distance is smaller than $\delta$, we can represent the graph using a sparse matrix. This improves the space complexity to $O(|E|)$ where $E$ is the set of edges in the graph. While $O(|E|)$ is still $O(n^2)$ is the worst-case, in practice, it tends to be much smaller, and results in manageable space complexity.
>
> Concretely, on our largest dataset, ImageNet with DINO embedding ($n=1281167$, $d=384$) ProbCover takes a little under $25$ minutes using a single NVIDIA A4000 GPU, with total memory consumption of $12GB$ ($\delta=0.55$).
>
> We added a more elaborated discussion on this important matter to the paper, in Appendix F, with a reference to lines 172-173 in the main paper.
>
> **ImageNet evaluation**
>
> Indeed, the ability to *scale* active learning methods to large datasets is very important. Following your request, we conducted experiments on the full ImageNet dataset. Using a budget of 1000 samples, we compared ProbCover (with $\delta=0.55$ and DINO features), TypiClust (with DINO features), random selection, margin, and coreset.
>
> Our results are qualitatively similar to the other experiments reported in the original submission, once again revealing very significant benefits to TypiClust and ProbCover as compared with random selection. Importantly, ProbCover improves over TypiClust. Other methods are on par with random selection. These results are now integrated into the new revision, shown in Figs. 2,3,4
>
> We note that we could not scale BADGE to ImageNet, as it requires storing a matrix of $n\times w\times C$ of hallucinated gradients, where $w=512$ and $C=1000$ is the number of classes. This matrix takes 2.27TB of memory using float32. In contrast, ProbCover is characterized by a fairly fast and memory-efficient  sample selection. We note that ProbCover calculation happens only as a pre-processing stage to the training of the network, hence the running time of the entire training does not go up dramatically.
>
> **Fig.3(a) and Fig.4(a) of TypiClust**
>
> Both TypiClust and ProbCover use representation learning to learn a meaningful embedding space for the data. In TypiClust, they used $2$ different representation spaces: one based on SimCLR (denoted as $TCP_{RP}$) and another based on SCAN (denoted as $TCP_{DC}$). We reported results of ProbCover on the SimCLR representation only, as it was shown to yield better results on most datasets and budgets. Therefore, we only compared ProbCover and TypiClust on this representation, to eliminate sources of confounds in our results. We clarified this point in the current revision, in lines 216-218 and in Appendix E.3
>
> For completeness, we ran ProbCover on CIFAR-10 using the SCAN representation (similarly to Fig.4(a) of TypiClust) as well. We found that similar to the results using SimCLR, ProbCover reaches higher accuracy than TypiClust on CIFAR-10 when using this representation as well. This experiment is described in the new revision in Appendix E.3, and its results can be found in Fig.12

---

### Meta-Review · Area_Chair_LWw9 · 2022-08-26

**Recommendation:** Accept
**Confidence:** Certain

**Metareview:**

This paper introduces an active learning method, ProbCover, that seeks to maximize probability coverage for the low budget regime. It also provides theoretical analysis and a dual way to view the proposed method with respect to methods better suited to the high budget regime like Coreset. The paper received two weak accept and a weak reject rating. After reviewing the paper, reviews, author responses, add additional discussion with reviewers, I believe that on the balance the strengths of the paper outweigh the weaknesses. Reviewers overall appreciated the importance of the task, the theoretical analysis, and thorough experiments. Most of the questions and clarifications about the approach were sufficiently addressed in the rebuttal, as well as additional requested experiments. Reviewers considered the use of a representation space from an auxiliary model to be a weakness, but I agree with the assessment (also concurred by reviewer) that this is not a limitation that invalidates the contributions of the paper. In practice it is a reasonable approach given the low budget regime. Reviewers also questioned the comparison with TypiClust, which was originally performed using only the SimCLR representation space, and not the one based on SCAN that TypiClust also used. The authors added comparison based on the SCAN representation for CIFAR-10 in their rebuttal, which is appreciated, and showed improvement in this setting as well. However, since an experiment on only one dataset was provided, and the improvement is smaller than for the main reported results using SimCLR (and TypiClust outperforms ProbCover at the smallest budget), the authors are highly encouraged include more comprehensive comparisons using the SCAN representation as well in the final paper for completeness. Overall, acceptance is recommended for this paper.

**Award:**

No

---

### Decision · Program_Chairs · 2022-09-14

Accept